# CD8+ T cell-derived Fgl2 regulates immunity in a cell-autonomous manner via ligation of FcγRIIB

Kelsey B. Bennion [1,2,3], Danya Liu[1], Abdelhameed S. Dawood[4,5], Megan M. Wyatt [1,2,3,6], Katie L. Alexander [1,7], Mohamed S. Abdel-Hakeem[4,5], Chrystal M. Paulos [1,2,3,6] & Mandy L. Ford [1,2,3,7] ✉

The regulatory circuits dictating CD8+ T cell responsiveness versus exhaustion during anti-tumor immunity are incompletely understood. Here we report that tumor-infiltrating antigen-specific PD-1+ TCF-1– CD8+ T cells express the immunosuppressive cytokine Fgl2. Conditional deletion of *Fgl2* specifically in mouse antigen-specific CD8+ T cells prolongs CD8+ T cell persistence, suppresses phenotypic and transcriptomic signatures of T cell exhaustion, and improves control of the tumor. In a mouse model of chronic viral infection, PD-1+ CD8+ T cell-derived Fgl2 also negatively regulates virus-specific T cell responses. In humans, CD8+ T cell-derived Fgl2 is associated with poorer survival in patients with melanoma. Mechanistically, the dampened responsiveness of WT *Fgl2*-expressing CD8+ T cells, when compared to *Fgl2*-deficient CD8+ T cells, is underpinned by the cell-intrinsic interaction of Fgl2 with CD8+ T cell-expressed FcγRIIB and concomitant caspase 3/7-mediated apoptosis. Our results thus illuminate a cell-autonomous regulatory axis by which PD-1+ CD8+ T cells both express the receptor and secrete its ligand in order to mediate suppression of anti-tumor and anti-viral immunity.

CD8+ T cell responses are a critical component of immunity to cancer and chronic viral infections[1–3]. Persistent antigen exposure under these conditions is a driving factor in scarring T cells: impairing their ability to differentiate into long-lived memory cells and compromising their effector function[4–12], resulting in a state of T cell exhaustion. Seminal studies have demonstrated that CD8+ T cell exhaustion results in failed immunosurveillance in the context of cancer, as well as viral persistence in the context of infection, both contributing to disease progression. Exhausted CD8+ T cells exhibit impaired proliferative capacity, reduced effector function, and eventually undergo apoptosis[13–15]. However, the mechanisms via which this occurs are incompletely understood. In this study, we found that under chronic

inflammatory conditions, antigen-specific PD-1+ CD8+ T cells can secrete the immunoregulatory cytokine fibrinogen-like protein 2 (Fgl2) that regulates the antigen-specific CD8+ T cell response to tumor and virus in a cell-autonomous manner.

Although the expression of Fc receptors on T cells was originally suggested in the 1970s, dogma over the past few decades has held that T cells do not express Fc receptors. We previously showed that FcγRIIB, the only inhibitory IgG-Fc receptor, is expressed on a subset of highly differentiated effector CD8+ T cells in mice and humans[16–19]. However, the mechanisms by which FcγRIIB controls CD8+ T cells were incompletely elucidated. Here we show that FcγRIIB-expressing CD8+ T cells are "poised" for deletion, but do not undergo deletion until the

1Department of Surgery, Emory University School of Medicine, Atlanta, GA, USA. 2Emory Winship Cancer Institute, Atlanta, GA, USA. 3Cancer Biology PhD Program, Emory University, Atlanta, GA, USA. 4Emory Vaccine Center, Emory University School of Medicine, Atlanta, GA, USA. 5Pathology Advanced Translational Research Unit (PATRU), Department of Pathology and Laboratory Medicine, Emory University School of Medicine, Atlanta, GA, USA. 6Department of Microbiology and Immunology, Emory University, Atlanta, GA, USA. 7Immunology and Molecular Pathogenesis PhD Program, Emory University, Atlanta, GA, USA. ✉e-mail: mandy.ford@emory.edu

cell is driven to also express Fgl2, which then functions in an auto-feedback loop to induce apoptosis in a cell-autonomous manner. These data provide insight into a mechanism whereby exhausted T cells play an active role in their own deletion and establish the paradigm that "exhausted" T cells are not functionally inert but actively participate in the resolution of an immune response by inducing deletion of antigen-specific CD8+ T cells via the Fgl2/ FcγRIIB axis. In this way, the differentiation of Fgl2-expressing CD8+ T cells may contribute to the prevention of immune pathology and the return to immune homoeostasis.

## Results

### Fgl2 expression in CD8+ T cells is tightly associated with exhaustion and portends increased mortality in human patients with melanoma

To interrogate characteristics of tumor-infiltrating CD8+ T cells that promote T cell exhaustion in patients with cancer, single-cell RNA-sequencing data of dissociated cells isolated from squamous cell carcinomas were analyzed. Results indicated that *Fgl2* RNA was present in tumors (Supplementary Fig. 1a, b), and was significantly increased in activated CD8+ TIL compared to naive CD8+ TIL ($p < 0.0001$). At the protein level, Fgl2 cytokine was induced in CD8+ TIL from digested melanoma patient tumor tissue upon stimulation with anti-CD3/CD28 (Supplementary Fig. 1c, d). Likewise, a significantly higher frequency and mean fluorescence intensity (MFI) of Fgl2 cytokine expression was observed within CD44hi CD8+ T cells compared to CD44lo CD8+ T cells during ex vivo peptide stimulation of cells isolated from the spleen ($p < 0.01$) and tumor ($p < 0.05$) of B16-OVA challenged mice (Supplementary Fig. 1e–g). Additionally, we observed a significant increase in the MFI of Fgl2 and frequency of Fgl2 producers among OVA-specific CD8+ T cells during a multi-day stimulation with cognate antigen compared to unstimulated controls ($p < 0.05$) (Supplementary Fig. 1h–k).

To dissect the characteristics of Fgl2-expressing activated CD8+ T cells, mice were challenged with B16 tumors, and Fgl2 was measured on CD8+ T cells isolated from the blood, draining lymph node, spleen, and tumor via flow cytometry (Fig. 1a). Fgl2-expressing CD8+ T cells were contained almost entirely ($p < 0.001$, Fig. 1b) within the population of cells that express PD-1, a cell surface receptor that marks antigen-experienced and functionally exhausted CD8+ T cells in the context of cancer and chronic viral infection. Fgl2+ CD8+ T cells were also less proliferative than their Fgl2− counterparts as measured by Ki-67 (Fig. 1c). Relative to Fgl2− CD8+ T cells, Fgl2+ CD8+ T cells exhibited significantly reduced expression of the transcription factors TCF-1 and Eomes (Fig. 1d, e), both of which have been associated with stem-like PD-1+ T cells[20–24]. These data show that Fgl2 expression in CD8+ T cells is tightly associated with a more highly differentiated, exhausted-like CD8+ T cell status.

The upregulation of Fgl2 expression in exhausted CD8+ T cells was externally validated using the publicly available single-cell RNA sequencing dataset published by Carmona et al.[25]. in a B16 murine melanoma model using the single-cell analysis software, BBrowser 2[26]. Upon reanalysis of 7,174 cells, we found that the user-classified PD-1 intermediate (PD-1int) effector population (characterized by low expression of inhibitory receptors but moderate expression of cytotoxic molecules) and PD-1 high (PD-1hi) exhausted (characterized by high inhibitory receptor and high cytotoxic molecules expression) tumor-infiltrating CD8+ T cells express significantly more *Fgl2* than naive (characterized by high levels of *Tcf7*, *Sell*, and low *CD44* expression) CD8+ T cells at the tumor (Fig. 1f). The tSNE is also shown for naive (Fig. 1g), PD-1int (Fig. 1h), and PD-1hi (Fig. 1i) CD8+ within the tumor. A signature based on genes well known for their expression on antigen-experienced and/or exhausted cells, *Pdcd1* and *Havcr2* revealed linked expression of *Fgl2* in both PD-1int effector and PD-1hi exhausted CD8+ T cell populations.

Given this finding of Fgl2 production by exhausted PD-1+ CD8+ T cells in both humans and mice with cancer, we next sought to determine the association of CD8+ T cell-expressed Fgl2 with patient outcomes in melanoma. The expression of *Fgl2* on CD8+ TIL in melanoma patients treated with immune checkpoint therapy was interrogated using the single cell RNA sequencing dataset published by Sade-Feldman et al.[27] (Fig. 1j). Although regulatory T cells (Tregs) and macrophages have both been identified as cellular sources of Fgl2[28–30], expression of *Fgl2* in neither Tregs nor macrophages correlated with decreased patient survival post-immune checkpoint therapy (Fig. 1k, l)[27]. In contrast, however, increased expression of *Fgl2* on exhausted CD8+ T cells was significantly correlated with decreased patient survival (Fig. 1m)[27]. Together these data demonstrate that Fgl2 is produced by CD44hi antigen-specific CD8+ T cells upon activation (~5%) (Supplementary Fig. 1), but that it is the PD-1+ exhausted-like antigen-specific CD8+ T cells that produce the most Fgl2 (~40%) (Fig. 1). Further, exhausted CD8+ T cell-derived Fgl2 is associated with worsened mortality in human patients with melanoma.

### Conditional deletion of Fgl2 from tumor-specific CD8+ T cells increases their persistence and improves tumor control

To elucidate the role of autonomously-produced Fgl2 in the development of exhaustion programs in CD8+ T cells, WT vs. *Fgl2*-deficient OVA-specific CD8+ OT-I T cells were adoptively transferred into *Fgl2−/−* hosts that were then challenged with B16-OVA (Fig. 2a). By rendering the host *Fgl2* deficient, we were able to isolate the impact of Fgl2 production from the tumor-specific CD8+ T cells. Results indicated that recipients of *Fgl2−/−* tumor-specific CD8+ T cells exhibited significantly decreased B16 melanoma tumor size as compared to recipients of WT tumor-specific CD8+ T cells (Fig. 2b). Moreover, both the frequencies and absolute numbers of *Fgl2−/−* tumor-specific CD8+ T cells were significantly increased relative to WT tumor-specific CD8+ T cells at day 14 in the draining lymph node (Fig. 2c–e) and spleen (Fig. 2f, g) of tumor-challenged mice. To more fully elucidate differences between WT vs. *Fgl2−/−* tumor-specific CD8+ T cells, OT-I were isolated from the spleens of day-14 B16-OVA challenged mice and sorted for bulk RNA-sequencing followed by downstream analysis of differentially expressed genes through Gene Set Enrichment Analysis (GSEA). We found that apoptosis-related genes were enriched in WT vs. *Fgl2−/−* tumor-specific CD8+ T cells (Fig. 2h). Of note, these pathways were not differentially expressed in the comparison analyses of naive WT vs. *Fgl2−/−* CD8+ T cells. These differences in apoptosis-related genes suggest that increased accumulation of *Fgl2−/−* tumor-specific T cells is underpinned by a difference in cell death. To understand if genetic deletion of *Fgl2* increased the persistence of tumor-specific CD8+ T cells by inhibiting apoptosis, active cleaved caspase 3/7 was measured in WT vs. *Fgl2−/−* tumor-specific CD8+ T cells. We observed an increased frequency of caspase 3/7+ tumor-specific CD8+ T cells among WT CD8+ T cells (Fig. 2i, j) compared to *Fgl2−/−* CD8+ T cells ($p < 0.05$). The decreased cell death in *Fgl2−/−* CD8+ T cells was accompanied with a concomitant decrease in exhaustion, in that genes associated with a more memory and less exhausted CD8+ T cell phenotype were enriched in *Fgl2−/−* OT-I compared to WT OT-I (Fig. 2k). Further, *Fgl2−/−* tumor-specific CD8+ T cells exhibited significantly decreased frequencies of PD1+ TIM3+−expressing cells (Fig. 2l–n) ($p < 0.01$). When WT and *Fgl2−/−* tumor-specific CD8+ T cells were co-adoptively transferred (Fig. 2o), *Fgl2−/−* tumor-specific CD8+ T cells were able to outcompete WT tumor-specific CD8+ T cells (Fig. 2p) in the lymph node (Fig. 2q) and in the tumor (Fig. 2r). Taken together, these data demonstrate that Fgl2 produced by antigen-specific CD8+ T cells functioned to suppress the persistence of tumor-specific CD8+ T cells in a cell-intrinsic manner, likely through induction of apoptosis, and rendered them less able to limit tumor growth in a murine model of melanoma.

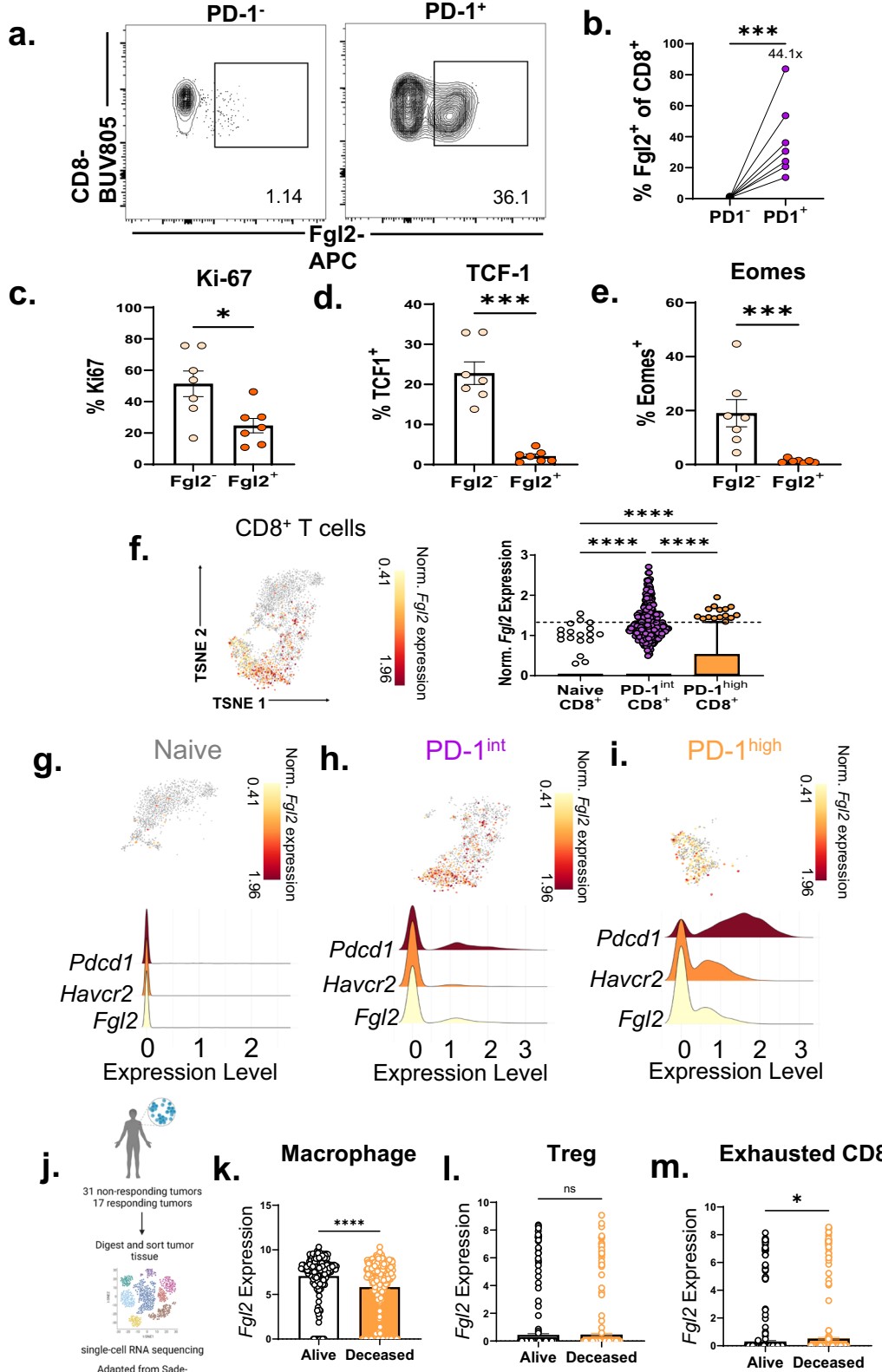

## Fgl2 is increased in exhausted P14 CD8+ T cells during LCMV Clone 13 infection, conditional deletion from virus-specific CD8+ T cells increases persistence and decreases exhaustion

To generalize these findings to another model of chronic inflammation and to determine if $Fgl2^{-/-}$ antigen-specific cells persisted longer because of increased expansion or impaired contraction, the role of CD8+ T cell-expressed Fgl2 was investigated in the lymphocytic choriomeningitis virus clone 13 model of chronic viral infection. Given the transcriptomic similarities of exhausted CD8+ T cells in models of chronic viral infection and cancer[31], we queried the expression of *Fgl2* in the defined terminally exhausted CD8+ T cell subset by Miller et al. [31] in B16-OVA and LCMV-Clone 13. We found that *Fgl2* is a differentially expressed gene upregulated in the terminally exhausted vs. progenitor exhausted CD8+ T cells in both

**Fig. 1 | Fgl2 expression in CD8+ T cells is tightly associated with exhaustion and portends increased mortality in human patients with melanoma.**
**a** Representative flow plots and **b** summary data showing frequency of Fgl2+ cells within PD-1− vs. PD-1+ CD8+ T cells from mice fourteen days post B16F10 melanoma challenge (p = 0.0006). Summary data showing frequency of **c** Ki-67+ (p = 0.0256), **d** TCF-1+ (p = 0.0006), and **e** Eomes+ (p = 0.0006) cells within Fgl2− vs. Fgl2+ CD8+ T cells from B16-challenged mice. Representative data from two independent experiments, n = 7 mice per group. **f** Summary data and tSNE plot from a publicly available single cell RNA-sequencing dataset deposited by Carmona et al. visualizing Fgl2 gene expression of CD8+ tumor-infiltrating lymphocytes (TIL) isolated from B16F10 challenged mice (n = 7). t-SNE visualization and accompanying comparison of exhausted-like gene signature (Pdcd1 and Havcr2) with Fgl2 expression on **g** naive CD8+ T cells (n = 1249 cells), **h** PD-1int effector CD8+ (n = 1548 cells), and **i** PD-

1high exhausted CD8+ T cells (n = 588 cells). For each comparison, p < 0.0001. **j** Schematic and summary data of Fgl2 expression on **k** macrophages (p < 0.0001), **l** regulatory T cells (Treg) (p = 0.6819), and **m** exhausted CD8+ T cells (p = 0.048) correlated to patient survival from a publicly available dataset deposited by Sade-Feldman et al. consisting of 16,291 single cell transcriptome profiles from patient tumors (n = 32 patients), data was normalized to housekeeping gene expression within each cell is shown. Mann-Whitney non-parametric, unpaired two-sided tests was used when comparing two groups, Kruskall-Wallis non-parametric, one-way ANOVA with Dunn's multiple comparisons test was used when comparing >2 groups. The error bar in summary figures denotes mean ± SEM. *p < 0.05 ***p < 0.001, ****p < 0.0001. Source data are provided as a Source Data file. **j** was created with BioRender.com released under a Creative Commons Attribution-NonCommercial-NoDerivs 4.0 International license.

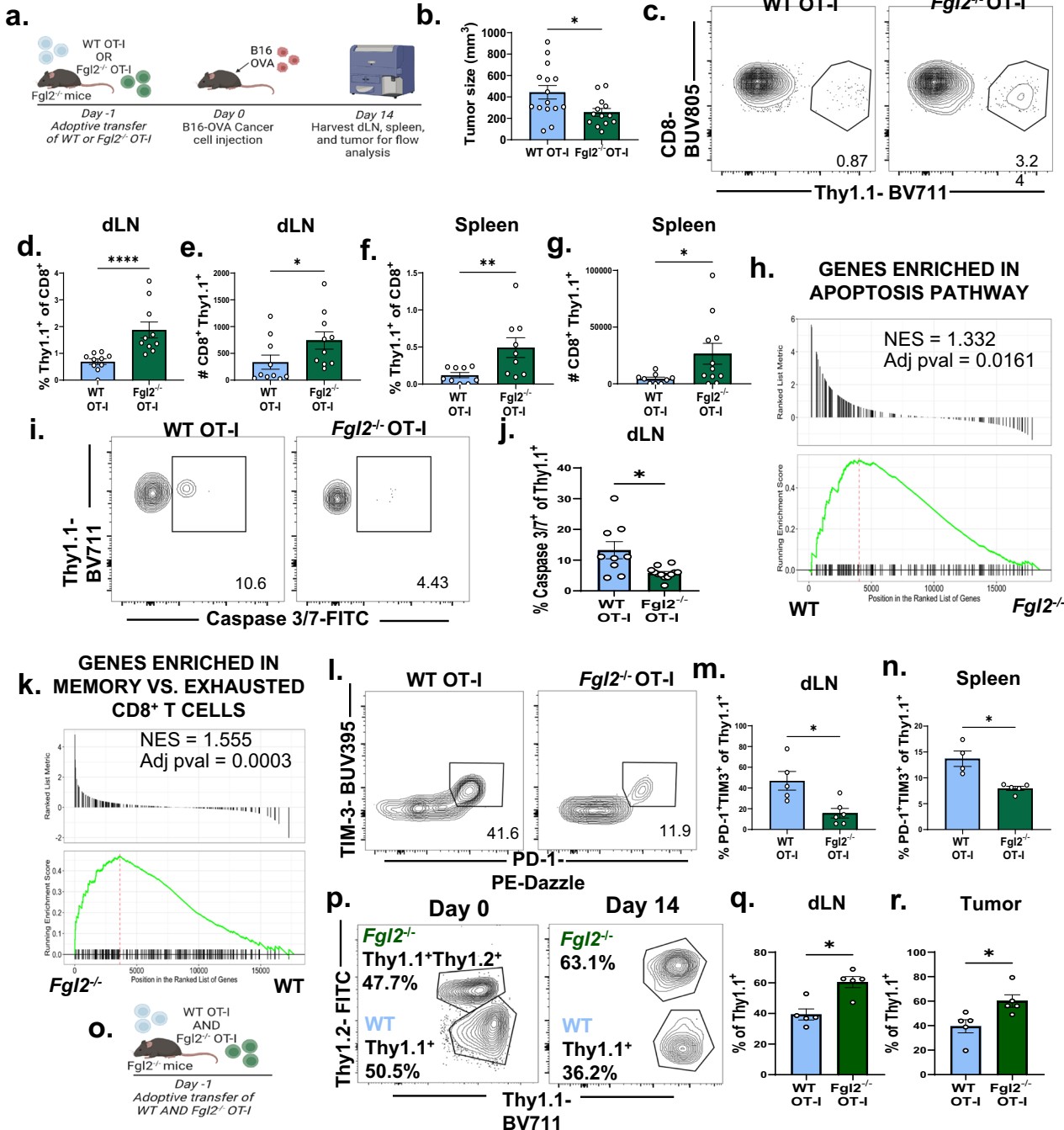

**Fig. 2 | Conditional deletion of *Fgl2* from tumor-specific CD8⁺ T cells increases their persistence and improves tumor control. a** Schematic wherein *Fgl2⁻/⁻* mice were given WT or *Fgl2⁻/⁻* OCD8⁺ CD8⁺ T cells one day prior to cancer cell inoculation. Fourteen days later, immune cell function was measured via flow cytometry and bulk RNA-sequencing. **b** Tumor size as measured on day fourteen post tumor challenge, pooled data from three independent experiments, *n* = 15 mice in WT OT-I group and 13-15 mice in *Fgl2⁻/⁻* OT-I group (*p* = 0.0253). **c** Representative flow plots and summary data showing **d** frequency and **e** cell count of CD8⁺Thy1.1⁺ cells present in the draining lymph node of mice given WT vs. *Fgl2⁻/⁻* OT-I. Pooled data from two independent experiments, *n* = 10 mice per group, (*p* < 0.0001) and *p* = 0.0288, respectively). **f** Frequency and **g** cell count in the spleen of mice is also shown. Pooled data from two independent experiments, *n* = 9 in the WT OT-I group and 11 mice in the *Fgl2⁻/⁻* OT-I group (*p* = 0.0097 and *p* = 0.0308, respectively). **h** Gene set enrichment analysis of differentially expressed genes found by bulk RNA-sequencing of WT vs. *Fgl2⁻/⁻* OT-I (isolated day 14 post tumor challenge) revealed that WTCD8⁺ D8⁺ D8⁺ T cells were significantly enriched for genes in the HALLMARK_APOPTOSIS MSigDB pathway compared *Fgl2⁻/⁻* OT-I (*n* = 4 mice per group) with a normalized enrichment score (NES) = 1.332 and an adjusted *p* value = 0.0161. **i** Representative flow plots and **j** summary data showing the frequency of cleaved caspase 3/7⁺ WT and *Fgl2⁻/⁻* OT-I, pooled data from two

independent experiments (*n* = 9 in the WT OT-I group and 12 mice in the *Fgl2⁻/⁻* OT-I group), (*p* = 0.0218). **k** Gene set enrichment analysis of differentially expressed genes revealed that *Fgl2⁻/⁻* OT-I were significantly enriched for genes upregulated in memory compared to exhausted CD8⁺ T cells isolated day 30 post LCMV-Clone 13 infection compared WT OT-I (*n* = 4 mice per group) with a normalized enrichment score (NES) = 1.555 and an adjusted *p* value = 0.0003. **l** Representative flow plots and summary data in the **m** draining lymph node (*n* = 5 in the WT OT-I group and 6 mice in the *Fgl2⁻/⁻* OT-I group) and **n** spleen (*n* = 4 in the WT OT-I group and 5 mice in the *Fgl2⁻/⁻* OT-I group) showing frequency of PD-1⁺TIM-3⁺ OT-I within WT and *Fgl2⁻/⁻* OT-I populations, (*p* = 0.0173 and *p* = 0.0159, respectively). Representative data from one of two independent repeats. **o** Schematic showing co-adoptive transfer wherein WT and *Fgl2⁻/⁻* OT-I were transferred at a 1:1 ratio and mice were subsequently challenged with B16-OVA. **p** Representative flow plots gated on Thy1.1⁺ and summary data in the**Q** dLN ad**(R)** tumor of WT (Thy1.1⁺) vs. *Fgl2⁻/⁻* (Thy1.1⁺Thy1.2⁺) in tumor mice, *n* = 5 mice per group (*p* = 0.0159 and *p* = 0.0159, respectively). Mann-Whitney non-parametric, unpaired two-sided tests were used. The error bar in summary figures denotes mean ± SEM. \**p* < 0.05 \*\**p* < 0.01, \*\*\*\**p* < 0.0001. Source data are provided as a Source Data file. **a**, **o** is created with BioRender.com released under a Creative Commons Attribution-NonCommercial-NoDerivs 4.0 International license.

models (Fold Change (FC) = -3.57, adjusted *p*-value = 6.67e-59 and FC = -3.70, adjusted *p*-value = 2.83e-57, respectively (Fig. 3a). In an analysis of publicly available data comparing exhausted vs. naive P14 CD8⁺ T cells isolated day 45 after LCMV-clone 13 infection deposited by Im et al. [23], the *Fgl2* gene was significantly increased in exhausted P14 vs. naive P14 (FC = 8.425, log₁₀pvalue = 6.873) (Fig. 3b). *Pdcd1, Havcr2, and GzmK* genes are also shown as a frame of reference (FC = 7.836, log10pvalue = 6.586 and FC = 7.672, log₁₀pvalue = 9.387, respectively). Given these data, *Fgl2⁻/⁻* LCMV antigen-specific CD8⁺ P14 T cells were generated to measure the impact of antigen-specific CD8⁺–derived Fgl2 in the LCMV-Clone 13 model. *Fgl2⁻/⁻* recipients of either WT P14 or *Fgl2⁻/⁻* P14 were infected with LCMV cl-13; blood was sampled on days 8, 12, and 15 and groups of animals were sacrificed on days 15 post-infection (Fig. 3c). While WT and *Fgl2⁻/⁻* P14 exhibited similar expansion on day 8, *Fgl2⁻/⁻* P14 exhibited increased persistence in blood (Fig. 3d, e), lymph node (Fig. 3f) and spleen (Fig. 3g) relative to WT P14 T cells (*p* < 0.01) on days 12 and 15 post-infection. Moreover, the frequency of PD-1⁺TIM-3⁺ terminally-exhausted-like WT P14 increased from d8 to d15, while the frequency of PD-1⁺TIM-3⁺ cells among *Fgl2⁻/⁻* P14 decreased from d8 to d15 (*p* < 0.05) (Fig. 3h, i). These data demonstrate that Fgl2 regulates the antigen-specific CD8⁺ T cell response to chronic infection in a CD8⁺ T-cell autonomous manner.

## Genetic deletion of Fgl2 from antigen-specific CD8⁺ rescues FcγRIIB⁺ CD8⁺ T cells in cancer and chronic viral infection

We next sought to address the mechanism by which Fgl2 regulates CD8⁺ T cell immunity. Fgl2 in its soluble form is known to bind to the inhibitory Fc receptor FcγRIIB on B cells and dendritic cells[32,33]. Moreover, we recently showed that cell-autonomous expression of FcγRIIB regulates CD8⁺ T cell responses in the context of tumor and virus[16–19]. Thus, we queried if the immunoregulatory impact of Fgl2 produced by exhausted antigen-specific CD8⁺ occurred via FcγRIIB.

To understand if Fgl2 produced by CD8⁺ T cells regulated FcγRIIB⁺ Ag-specific CD8⁺ T cells, we utilized the B16 model mentioned above in which WT vs. *Fgl2⁻/⁻* OT-I T cells were adoptively transferred into *Fgl2⁻/⁻* recipients which were challenged with B16-OVA (Fig. 4a, b). Results indicated that deletion of *Fgl2* from tumor-specific OT-I CD8⁺ T cells significantly increased both the frequency and cell count of FcγRIIB⁺ CD8⁺ OT-I in the draining lymph node (Fig. 4c–e). Likewise, during LCMV-clone 13 infection (Fig. 4f), the frequency and number of FcγRIIB⁺ cells were significantly increased among *Fgl2⁻/⁻* P14 as compared to among WT P14 in the blood (Fig. 4g, h), lymph node (Fig. 4i) and spleen (Fig. 4j). To further investigate the relationship between

FcγRIIB and Fgl2 expression, we revisited three RNA-sequencing datasets deposited by Wherry et al., Hudson et al. [24], and Doering et al.[34]. Results indicated that *Fcgr2b* is expressed in effector, memory, and transitory populations, but not in exhausted antigen-specific CD8⁺ T cells when *Fgl2* expression is highest (Supplementary Fig. 2a–d). Additionally, *Fcgr2b* was significantly less expressed in antigen-specific cells isolated from a chronic inflammatory environment (LCMV-Clone 13) compared to antigen-specific CD8⁺ isolated from an acute inflammatory environment (LCMV-Armstrong) (Supplementary Fig. 2e, f). Notably, *Fgl2* was significantly upregulated in cells isolated from LCMV-clone 13 compared to LCMV Armstrong-infected mice (Supplementary Fig. 2e, f). These data support the idea that Fgl2 produced by the antigen-specific CD8⁺ T cell population negatively regulates FcγRIIB-expressing CD8⁺ T cells, in that production of Fgl2 results in deletion of *Fcgr2b*-expressing cells. We next asked if Fgl2-mediated regulation of FcγRIIB-expressing CD8⁺ T cells occurred in an autocrine fashion. To answer this question, a co-adoptive transfer strategy was utilized wherein WT and *Fgl2⁻/⁻* OT-I were co-transferred at a 1:1 ratio (Fig. 4k) into the same recipient at day 0. Mice were then challenged with B16-OVA and sacrificed 14 days later. The frequencies of FcγRIIB⁺ cells among WT OT-I (Thy1.1⁺) or *Fgl2⁻/⁻* OT-I (Thy1.1⁺Thy1.2⁺) were measured (Fig. 4l). Results demonstrated increased frequencies of FcγRIIB⁺ *Fgl2⁻/⁻* OT-I vs. FcγRIIB⁺ WT OT-I (Fig. 4m). These data show that the regulation of FcγRIIB⁺ OT-I via Fgl2 occurs in an autocrine manner because the absence of Fgl2 from one OT-I population resulted in an increase in FcγRIIB⁺ cells in that population. Fgl2 does not seem to act in a paracrine manner as the secretion of Fgl2 bythe WT OT-I pool did not negatively impact the presence of FcγRIIB⁺ OT-I in the other OT-I population, the *Fgl2⁻/⁻* OT-I.

Because the induction of apoptosis is a mechanism by which Fgl2 regulates FcγRIIB⁺ B cells[32], we queried the frequency of caspase 3/7⁺ FcγRIIB⁺ vs FcγRIIB⁻ CD8⁺ T cells within the WT and *Fgl2⁻/⁻* tumor specific populations. While we did not observe a difference in caspase 3/7⁺ FcγRIIB⁻ tumor-specific CD8⁺ T cells (Fig. 4n, o), we found that a significantly higher frequency of FcγRIIB⁺ WT tumor-specific CD8⁺ T cells expressed caspase 3/7 compared to FcγRIIB⁺ *Fgl2⁻/⁻* deficient tumor-specific CD8⁺ T cells (*p* < 0.01) (Fig. 4p, q). Given the decrease in apoptotic FcγRIIB⁺ CD8⁺ T cells in *Fgl2⁻/⁻* cells, we next questioned whether Fgl2 was capable of binding and eliciting apoptosis of T cells via FcγRIIB.

## Fgl2 induces CD8⁺ T cell apoptosis via ligation of FcγRIIB in murine and human T cells

To begin to understand if Fgl acted to induce apoptosis of FcγRIIB⁺ Ag-specific CD8⁺ T cells, WT vs. *Fcgr2b⁻/⁻* Ag-specific murine CD8⁺

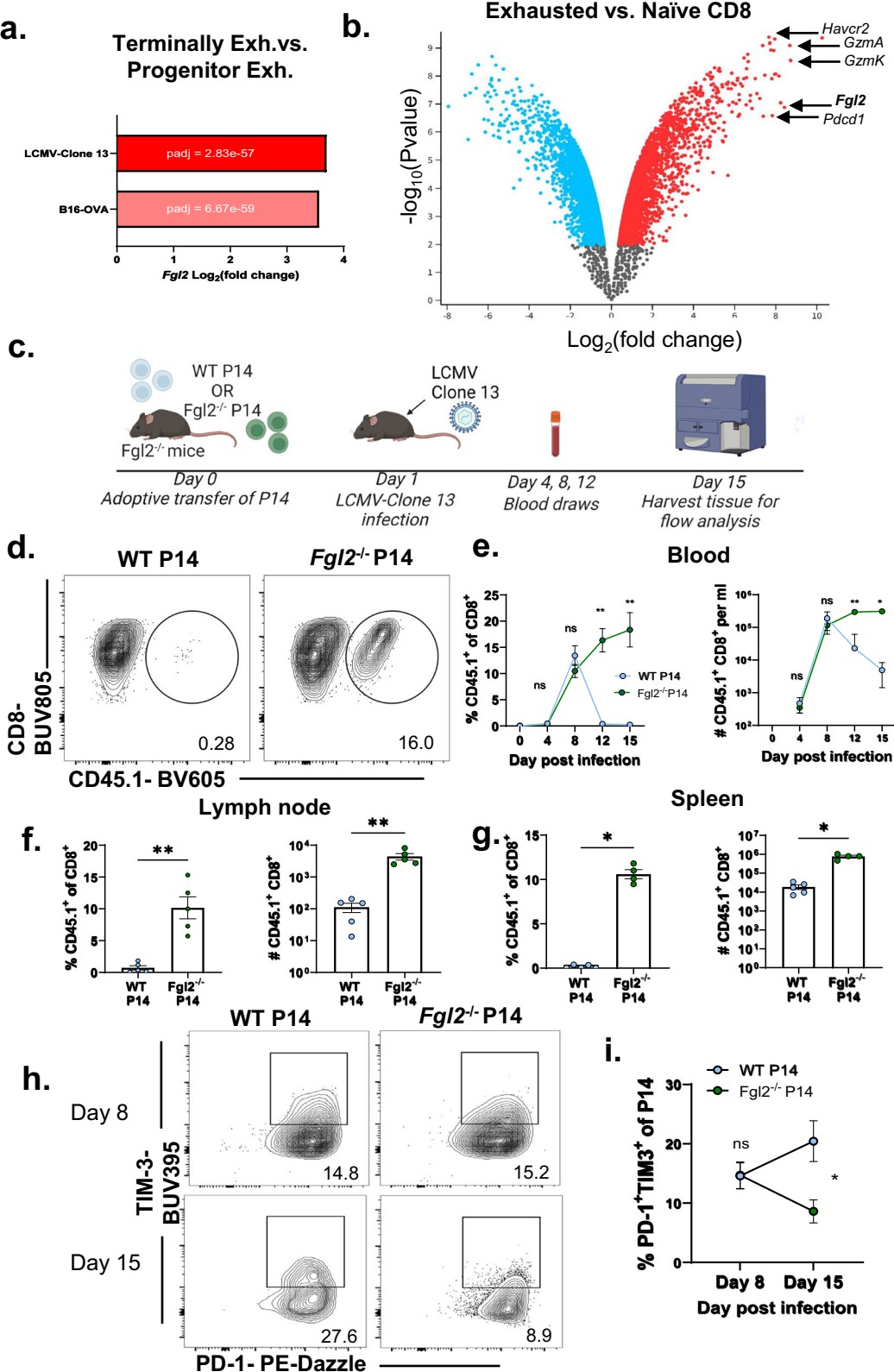

T cells were stimulated in the presence or absence of soluble Fgl2 in vitro (Fig. 5a). Results indicated that the addition of Fgl2 resulted in a significant increase in the frequency of active caspase 3/7⁺ 7-AAD⁺ cells among WT but not *Fcgr2b*⁻/⁻ antigen-specific CD8⁺ T cells (Fig. 5b, c). The ability of Fgl2 to induce FcγRIIB-dependent CD8⁺ T cell apoptosis in human cells was then assessed. Jurkat T cells were transfected with a plasmid encoding the *Fcgr2b* gene

along with a GFP tag (Fig. 5d). Surface expression of FcγRIIB protein was confirmed via flow cytometry (Fig. 5e). Next, *Fcgr2b*-transfected cells vs. mock transfected cells were incubated with Fgl2. Fgl2 binding to *Fcgr2b*-transfected cells was confirmed via flow cytometry, and could be abrogated by pre-incubation with anti-FcγRIIB. No Fgl2 binding was observed in the mock transfected cells (Fig. 5f). Lastly, transfected cells were stimulated with anti-CD3/28 for

**Fig. 3 | Fgl2 is increased in exhausted P14 CD8$^+$ T cells during LCMV Clone 13 infection, conditional deletion from virus-specific CD8$^+$ T cells increases persistence and decreases exhaustion. a** Bar graph showing *Fgl2* expression (log2 fold change) in antigen-specific CD8$^+$ T cells isolated from LCMV-Clone 13 ($n = 9,194$ cells, adjusted $p$ value = $2.83 \times 10^{-57}$) and B16-OVA ($n = 4313$ cells, adjusted $p$ value = $6.67 \times 10^{-59}$) in the publicly available single-cell RNA-sequencing dataset deposited by Miller et al. comparing progenitor exhausted vs. terminally exhausted CD8$^+$ T cells. **b** Volcano plot showing expression of genes upregulated in exhausted antigen-specific CD8$^+$ vs. naive CD8$^+$ T cells isolated on day 45 post LCMV-Clone 13 infection from the dataset deposited by Im et al., analyzed with GEO2R. **c** Schematic wherein *Fgl2$^{-/-}$* mice are given WT or *Fgl2$^{-/-}$* P14 one day prior to LCMV clone 13 infection. Mice were bled on days 4, 8, 12 and 15 and then sacrificed on day 15. **d** Representative flow plots and **e** time course of frequency ($p = 0.0095$ and $p = 0.0079$ for day 12 and day 15 comparisons respectively) and cell number ($p = 0.0043$ and $p = 0.0159$ for day 12 and day 15 comparisons respectively) per ml

of CD45.1$^+$ congenic marked WT (blue symbols) or *Fgl2$^{-/-}$* P14 (green symbols) within CD8$^+$ T cell subset in the blood ($n = 6$ mice per group). Frequency and cell number of P14 in the **f** lymph nodes ($n = 5$ mice per group). ($p = 0.0079$ and $p = 0.0079$ respectively) and **g** spleen in Clone 13-infected mice is also shown ($n = 5$ mice in WT P14 group and 4 mice in the *Fgl2$^{-/-}$* P14 group) ($p = 0.0159$ and $p = 0.0159$ respectively). **h** Representative flow plots and **i** summary data showing frequency of PD-1$^+$TIM-3$^+$ P14 in WT (blue symbols) vs. *Fgl2$^{-/-}$* P14 (green symbols) on day 8 vs. day 15 post infection ($n = 13$ mice in the WT P14 group and 10 mice in the *Fgl2$^{-/-}$* P14 group), $p = 0.0173$ for day 15 comparison). Representative data from two independent experiments. Mann-Whitney non-parametric, unpaired two-sided tests were used. The error bar, in summary, figures denotes mean ± SEM. **$p < 0.01$. Source data are provided as a Source Data file. **c** is created with BioRender.com released under a Creative Commons Attribution-NonCommercial-NoDerivs 4.0 International license.

24 hours in the presence of Fgl2 and stained for Annexin V and PI. Following incubation, significantly fewer GFP$^+$ cells were present in the Fgl2-treated compared to untreated wells, within the *Fcgr2b*-transfected but not mock transfected cultures ($p < 0.05$) (Fig. 5g). Moreover, Fgl2-treated wells exhibited a significant increase in the number of Annexin V$^+$ GFP$^+$ cells compared to untreated wells, within the *Fcgr2b*-transfected but not mock transfected cultures ($p < 0.05$) (Fig. 5h). These data demonstrate that Fgl2 binding to FcγRIIB on human CD8$^+$ T cells elicits the induction of apoptosis.

### FcγRIIB$^+$ T cell population contains a higher frequency of Fgl2-producers compared to FcγRIIB$^-$ CD8$^+$ T cells

After determining that Fgl2 does induce apoptosis of FcγRIIB-expressing T cells, we next sought to determine the dynamics of FcγRIIB and Fgl2 expression on CD8$^+$ T cells. Corroborating the finding that Fgl2 and FcγRIIB could be functioning to regulate the same cell pool, we previously noted that FcγRIIB$^+$ CD8$^+$ T cells express more *Fgl2* ($p = 0.057$) at the transcript level in RNA-sequencing data (Fig. 6a). Using the B16 melanoma model (Fig. 6b), we found that FcγRIIB$^+$ CD8$^+$ also express significantly more Fgl2 at the protein level than FcγRIIB$^-$ CD8$^+$ T cells (Fig. 6c–e). FcγRIIB$^+$ Fgl2$^+$ CD8$^+$ did not persist as we found significantly decreased frequencies of these double positive cells at day 21 compared to day 14 of tumor progression (Fig. 6f). This decrease could be due to a downregulation of either protein or the disappearance of these cells. Supportive of the latter hypothesis and our previous studies, we found that the double positive FcγRIIB$^+$ Fgl2$^+$ CD8$^+$ underwent more apoptosis compared to FcγRIIB$^+$ Fgl2$^-$ CD8$^+$ as measured by more caspase 3/7 and 7AAD staining (Fig. 6g). Upon transcription factor staining, we found that FcγRIIB$^+$ Fgl2$^+$ CD8$^+$ T cells were also less stem-like, less proliferative, and less memory-like, suggesting a differentiation trajectory wherein FcγRIIB$^+$ cells become FcγRIIB$^+$ Fgl2$^+$ as they become more differentiated (Fig. 6h–j). Upon investigation of the Sade-Feldman et al. dataset previously mentioned (Fig. 6k), we found that the Fgl2/FcγRIIB axis on CD8$^+$ was clinically relevant as elevated expression of both *Fcgr2b* and *Fgl2* on CD8$^+$ T cells in patients with melanoma correlated with increased resistance to immune checkpoint therapy (Fig. 6l).

### Discussion

Here we report that exhausted CD8$^+$ T cells can produce their own off switch in settings of cancer and chronic virus. These data are potentially paradigm-shifting in that they reveal an active role of exhausted T cells in mediating their own deletion. Thus, "exhausted" T cells are not functionally inert but actively participate in the resolution of an immune response by inducing deletion of antigen-specific T cells via the Fgl2/ FcγRIIB axis. In this way, the programmed differentiation of Fgl2-expressing "exhausted" T cells may contribute to the prevention of immunopathology and the return to immune homeostasis.

Of interest, this cell-autonomous pathway of CD8$^+$ T cell deletion is dependent on expression of the inhibitory receptor FcγRIIB. We show that antigen-specific CD8$^+$ cells bearing an exhaustion-related gene signature from mice as well as human TIL express robust levels of Fgl2, and this expression in melanoma patients is correlated with decreased patient survival during checkpoint inhibition. When challenged with B16 melanoma or LCMV-Clone 13, mice given Fgl2-deficient antigen-specific CD8$^+$ T cells exhibited enhanced antitumor and antiviral response via increased persistence of antigen-specific CD8$^+$ T cells. Given our previous work, we posited that this enhanced T cell response could be due to the lack of binding of Fgl2 to FcγRIIB$^+$ effector-like memory CD8$^+$ T cells. In contrast to long-held dogma, we and others have shown that FcγRIIB is expressed at the RNA and protein level on effector and memory CD8$^+$ T cells in the context of virus, cancer, and allograft[16–19,24,35–38]. Here, we mechanistically show that Fgl2 binds FcγRIIB and physically deletes FcγRIIB$^+$ CD8$^+$ T cells via apoptosis to consequently regulate the antigen-specific CD8$^+$ T cell response.

These data also demonstrate a cell-autonomous role for Fgl2 in limiting antigen-specific CD8$^+$ T cell responses. In contrast, previous studies have shown an indirect role of Fgl2 in suppressing T cell responses through inhibiting maturation of and decreasing antigen presentation by APCs[32,33,39–42]. The work presented herein is distinct in that we show a direct role of antigen-specific CD8$^+$ T cell-derived Fgl2 in limiting CD8$^+$ T cell responses. While previous studies have shown that Fgl2 can bind FcγRIIB on dendritic cells, macrophages, and B cells to induce apoptosis[32,43], data presented here show that Fgl2 can bind FcγRIIB on CD8$^+$ T cells in vitro[16]. In vivo and in vitro, in mice and human T cells, we more definitively show that binding of FcγRIIB by Fgl2 elicits apoptosis of FcγRIIB$^+$ CD8$^+$ T cells, forming a negative regulatory loop governing antigen-specific CD8$^+$ T cells. Our studies are corroborated by a recent report which showed that blocking Fgl2 in glioma results in a preservation of FcγRIIB$^+$ CD8$^+$ T cells in mice[44]. Together, these data bring to light a potential mechanism by which exhausted CD8$^+$ T cells actively dampen the T cell response via physical deletion of FcγRIIB$^+$ CD8$^+$ T cells. As memory CD8$^+$ T cells have been shown to express FcγRIIB, it is interesting to speculate that the deletion of FcγRIIB$^+$ CD8$^+$ T cells by Fgl2 may be a means by which the development of T cell exhaustion impairs the generation of memory T cells.

Our data corroborate the protumor role of Fgl2 in dysregulating the T-cell antitumor response but demonstrate a novel cellular source and mechanism by which Fgl2 dysregulates the antigen-specific response in a cell-intrinsic manner. CD8$^+$ T-cell derived Fgl2 is clinically relevant as we found that, surprisingly, *Fgl2* RNA expression on exhausted CD8$^+$ and not on Tregs or macrophages was correlated with decreased patient response to immune checkpoint therapy.

This work extends the work of other groups that have shown that Fgl2 has a protumor role in the context of glioma and hepatocellular carcinoma[29,30,40,44–46]. However, previous studies have focused on the

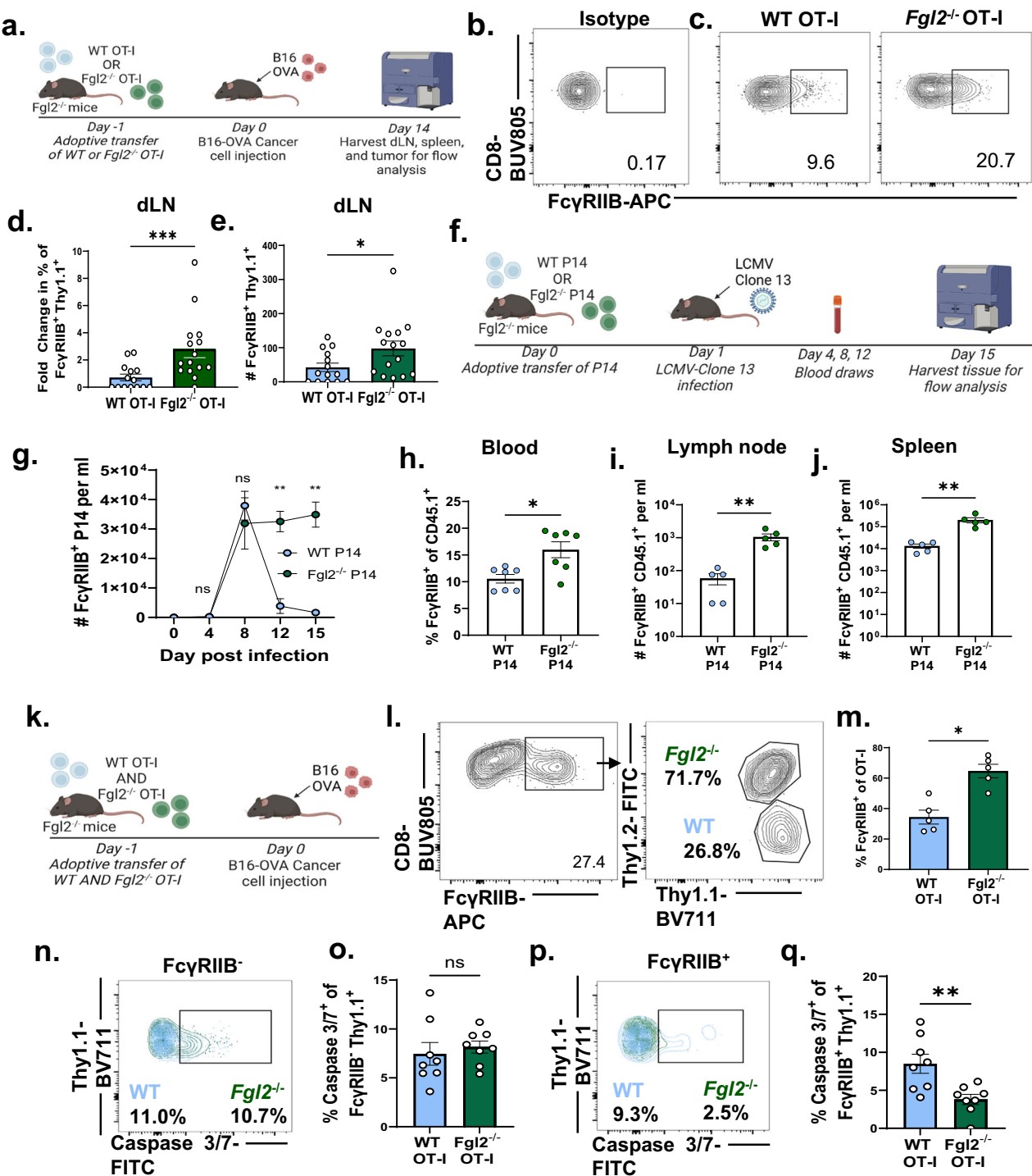

impact of Fgl2 produced from regulatory T cells, macrophages and tumor-associated cells (stroma, fibroblasts) as these cell types are known cellular sources of Fgl2 that can mediate immunosuppression on T cells[28–30,32,33,40–42,45–49]. As such, these studies showed the therapeutic efficacy of antibodies blocking Fgl2, suggesting that this approach may hold clinical promise in melanoma. However, as Fgl2 is not just an immunosuppressive cytokine, but a prothrombinase, the broad-scale manipulation of Fgl2 (e.g. Fgl2 blocking antibody described by others[44,50]) may have detrimental systemic effects as Fgl2 aids in coagulation, wound healing, and tolerance. These data build a compelling case that manipulation of Fgl2 from CD8[+] T cells either via TIL or CAR-T cell therapy could prolong the function and persistence of these cells without the systemic side effects of blocking Fgl2 from all

cellular sources. In summary, we have identified a regulatory signaling axis whereby CD8[+] T cells produce their own off switch and limits the CD8[+] T cell response to virus and tumor. The discovery of this pathway opens several potential avenues to manipulate the expression of the receptor and ligand, and signaling downstream of the receptor, to improve patient treatment options.

## Methods

### Mice

OT-I (Hogquist et al. [51] transgenic mice were purchased from Taconic Farms and bred to Thy1.1[+] (B6.PL-Thy1a/CyJ, Jackson Laboratory, Stock #000406) animals at Emory University. B6.Cg-Tcratm1Mom Tg(TcrLCMV)327Sdz/TacMmjax were purchased from Jackson

**Fig. 4 | Genetic deletion of *Fgl2* from antigen-specific CD8⁺ rescues FcγRIIB⁺ CD8⁺ T cells in cancer and chronic viral infection. a** Schematic wherein *Fgl2⁻/⁻* mice were given WT or *Fgl2⁻/⁻* OT-I CD8⁺ T cells one day prior to cancer cell inoculation. Fourteen days later, animals were sacrificed for immune cell function to be measured via flow cytometry. **b** Representative flow plot of the isotype control for FcγRIIB staining on CD8⁺ T cells. **c** Representative flow plots of FcγRIIB staining, gated prior on Thy1.1⁺ congenic marked OT-I. **d** Frequency of FcγRIIB⁺ OT-I within WT or *Fgl2⁻/⁻* OT-I subset normalized to WT OT-I ($p = 0.0009$) as well as **e** cell count of FcγRIIB⁺ OT-I ($p = 0.020$) in the draining lymph node of B16-challenged mice. Pooled data from three independent experiments, $n = 14$ in the WT OT-I group and 15 mice in the *Fgl2⁻/⁻* OT-I group). **f** Schematic wherein *Fgl2⁻/⁻* mice are given WT or *Fgl2⁻/⁻* P14 one day prior to LCMV clone 13 infection. Mice were bled on days 4, 8, 12 and 15 and then sacrificed on day 15. **g** time course of cell number per ml ($n = 6$ mice per group) ($p = 0.0043$ and $p = 0.0079$ for the day 12 and day 15 comparisons respectively) **h** and frequency at day 8 of FcγRIIB⁺ CD45.1⁺ congenic marked WT or *Fgl2⁻/⁻* P14 in the blood ($n = 7$ mice per group), $p = 0.0111$. Cell number of FcγRIIB⁺ P14 in the lymph nodes **i**, **j** spleen in Clone 13-infected mice is also shown ($n = 5$ mice per group), $p = 0.0079$ and $p = 0.0079$ respectively. **k** Schematic showing co-adoptive transfer wherein WT and *Fgl2⁻/⁻* OT-I were transferred at a 1:1 ratio and mice were subsequently challenged with B16-OVA. **l** Representative flow plots gated first on Thy1.1⁺, then FcγRIIB⁺, and then Thy1.1⁺ vs Thy1.2⁺ and **m** summary data in the dLN of FcγRIIB⁺ WT (Thy1.1⁺) vs. *Fgl2⁻/⁻* (Thy1.1⁺Thy1.2⁺) in tumor challenged mice, $n = 5$ mice per group, $p = 0.0159$. **n** Representative flow plot and **o** summary data showing frequency of Thy1.1⁺ caspase3/7⁺ of FcγRIIB⁻ WT vs. FcγRIIB⁻ *Fgl2⁻/⁻* OT-I ($n = 8$ mice per group). **p** Representative flow plot and **q** summary data showing frequency of Thy1.1⁺ caspase3/7⁺ of FcγRIIB⁺ WT vs. FcγRIIB⁺ *Fgl2⁻/⁻* OT-I, $n = 8$ mice per group, $p = 0.0070$. Pooled data from two independent experiments. Mann-Whitney non-parametric, unpaired two-sided tests were used. The error bar in summary figures denotes mean ± SEM. *$p < 0.05$, **$p < 0.01$, ***$p < 0.001$. Source data are provided as a Source Data file. **a**, **f**, **k** is created with BioRender.com released under a Creative Commons Attribution-NonCommercial-NoDerivs 4.0 International license.

---

Laboratory (Stock # 037394). EM:06078 Fcgr2b Fcgr2bB6null B6(Cg)-Fcgr2btm12Sjv/Cnbc (or *Fcgr2b⁻/⁻*) mice obtained under MTA with the Academisch Siekenhuis Leiden/Leiden University Medical Center and Dr. J.S. Verbeek. Cryopreserved embryos were shipped from the European Mutant Mouse Archive (EMMA) and re-derived at the Emory University Transgenic Mouse Core Facility. These mice were generated using embryonic stem cells from B6 mice. These *Fcgr2b⁻/⁻* mice made by Boross et al. [52] were bred to OT-I transgenic mice at Emory University. *Fgl2⁻/⁻* mice were a kind gift of Dr. Gary Levy[53], University of Toronto. *Fgl2⁻/⁻* mice were bred to OT-I or P14 Thy1.1⁺ TCR transgenic mice to generate *Fgl2⁻/⁻* OT-I mice and *Fgl2⁻/⁻* P14 mice. *Fgl2⁻/⁻* OT-I and *Fgl2⁻/⁻* P14 mouse strains were genotyped from tail biopsies using real time PCR with specific probes designed for each gene (Transnetyx). Unless otherwise indicated, mice were aged 8-10 weeks at the start of the experiment, experimental hosts were male and female, and donor animals for adoptive transfers were male or female. All experiments were performed under general anesthesia with maximum efforts made to minimize suffering. Euthanasia was performed using cervical dislocation after $CO_2$ asphyxiation. All animals were housed in specific pathogen-free animal facilities at Emory University. Control animals were bred separately in the same facility. Mice were housed at a humidity of ~55% and the dark/light cycle was 12 h/12 h.

## Tumor cell line culture and injection

The B16 melanoma cell line engineered to express the OVA epitope was provided by Dr. Yang-Xin Fu, University of Texas Southwest, Dallas, TX[54]. B16-OVA cells were cultured in RPMI 1640 (Sigma) and supplemented with 10% FBS, 1% P/S, 1% HEPES, 1% L-glutamine, and 0.05 MM 2-ME. Guidelines for B16 culture and cryopreservation outlined by the American Type Culture Collection were followed. Cancer cells were treated with 0.05% trypsin, washed with cold PBS, and filtered prior to cancer cell inoculation. 2-3x10⁵ B16-OVA cells were injected in cold PBS subcutaneously into the right flank according to the protocol established by Restifo et al. [55]. Tumor volume was monitored using electronic calipers and tumor size was calculated using the formula: tumor volume (mm³) = $(L × W^2)/2$, where L is the length and W is the width of the tumor. Animals were sacrificed when tumors reached IACUC endpoint (2 cm in either dimension). Cell lines were tested and found to be negative for mycoplasm as well as a myriad of other pathogens.

## Adoptive transfers

To monitor antigen-specific donor-reactive CD8⁺ T cell responses, 10⁶ OT-I transgenic T cells or 10³ P14 transgenic T cells were adoptively transferred through intravenous injection into naive mice 24 hours before tumor injection, or viral infection. For the adoptive transfer in B16-OVA tumor experiments, WT OT-I or *Fgl2⁻/⁻* OT-I transgenic T cells were harvested from the spleen of 8-10-week-old mice. For the adoptive transfer in LCMV-Clone 13 experiments, WT P14 or *Fgl2⁻/⁻* P14 transgenic T cells were isolated from the blood of 5-6 week old mice using Histopaque-1083 (Sigma). For co-adoptive transfer experiments, $5 × 10^5$ WT and $5 × 10^5$ *Fgl2⁻/⁻* OT-I were given at a 50–50% ratio, confirmed by flow cytometry the day before naive mice were challenged with tumor. In all adoptive transfers, cells were counted using a Nexcelom Cellometer (Nexcelom Bioscience) and stained with CD8-BV785, CD4-PacBlue, Thy1.1- PerCP, Vα2- FITC, and Vβ5- PE or Vβ8- PE (Biolegend). Frequency of OT-I cells was determined via CD8⁺ Thy.1.1⁺ Vα2 and Vβ5 TCR co-expression. Frequency of P14 cells was determined via CD8⁺ CD45.1⁺ Vα2 and Vβ8 TCR co-expression.

## RNA-Sequencing

WT or *Fgl2⁻/⁻* OT-I (Thy1.1⁺) were adoptively transferred into mice the day before mice were challenged with tumor. On day 14 post tumor challenge, splenocytes were homogenized and lymphocytes were isolated via Ficoll separation using Ficoll®-Paque PREMIUM 1.084 (Sigma). -10,000 CD3⁺CD8⁺Thy1.1⁺ cells were sorted, lysed, and then extracted using the RNAeasy Micro kit (Qiagen) with on-column DNase digestion. RNA quality was assessed using a Fragment Analyzer (Agilent) and then 1 ng of total RNA was used as input for cDNA synthesis using the Clontech SMART-Seq v4 Ultra Low Input RNA kit (Takara Bio) according to the manufacturer's instructions. Amplified cDNA was fragmented and appended with dual-indexed barcodes using the Nextera XT DNA Library Preparation kit (Illumina). Libraries were validated by capillary electrophoresis on a TapeStation 4200 (Agilent), pooled at equimolar concentrations, and sequenced with PE100 reads on an Illumina NovaSeq 6000, yielding -25 million reads per sample on average. Alignment was performed using STAR version 2.9.7a[56] and transcripts were annotated using GRCm38_102. Transcript abundance estimates were calculated internal to the STAR aligner using the algorithm of htseq-count[57]. Raw counts were analyzed in R using the DESeq2[58] version 1.41.12 and clusterProfiler[59] 4.8.2 packages from Bioconductor. Differentially expressed gene analyses were performed and differentially expressed genes were processed through GSEA. HALLMARK gene sets and C7 immunological signature gene sets from the Mouse Molecular Signatures Database (MSigDB) were used[34,60,61] Specifically, "HALLMARK_APOPTOSIS" and "GSE41867_MEMORY_VS_EXHAUSTED_CD8_TCELL_DAY30_LCMV_UP" are shown. A $p$-value $< 0.05$ adjusted with the Benjamini-Hochberg correction was considered significant. Adjusted $p$-value and enrichment scores are shown. No custom software that is not publicly available was utilized in this study. However, code is available upon request.

## LCMV-Clone 13 infection

A total of 5-8 week old WT and *Fgl2⁻/⁻* mice were infected with 4x10⁶ PFU of LCMV-Clone 13 via intravenous injection one day after adoptive transfer and housed in BSL2 facilities. LCMV-Clone 13 was a kind gift of

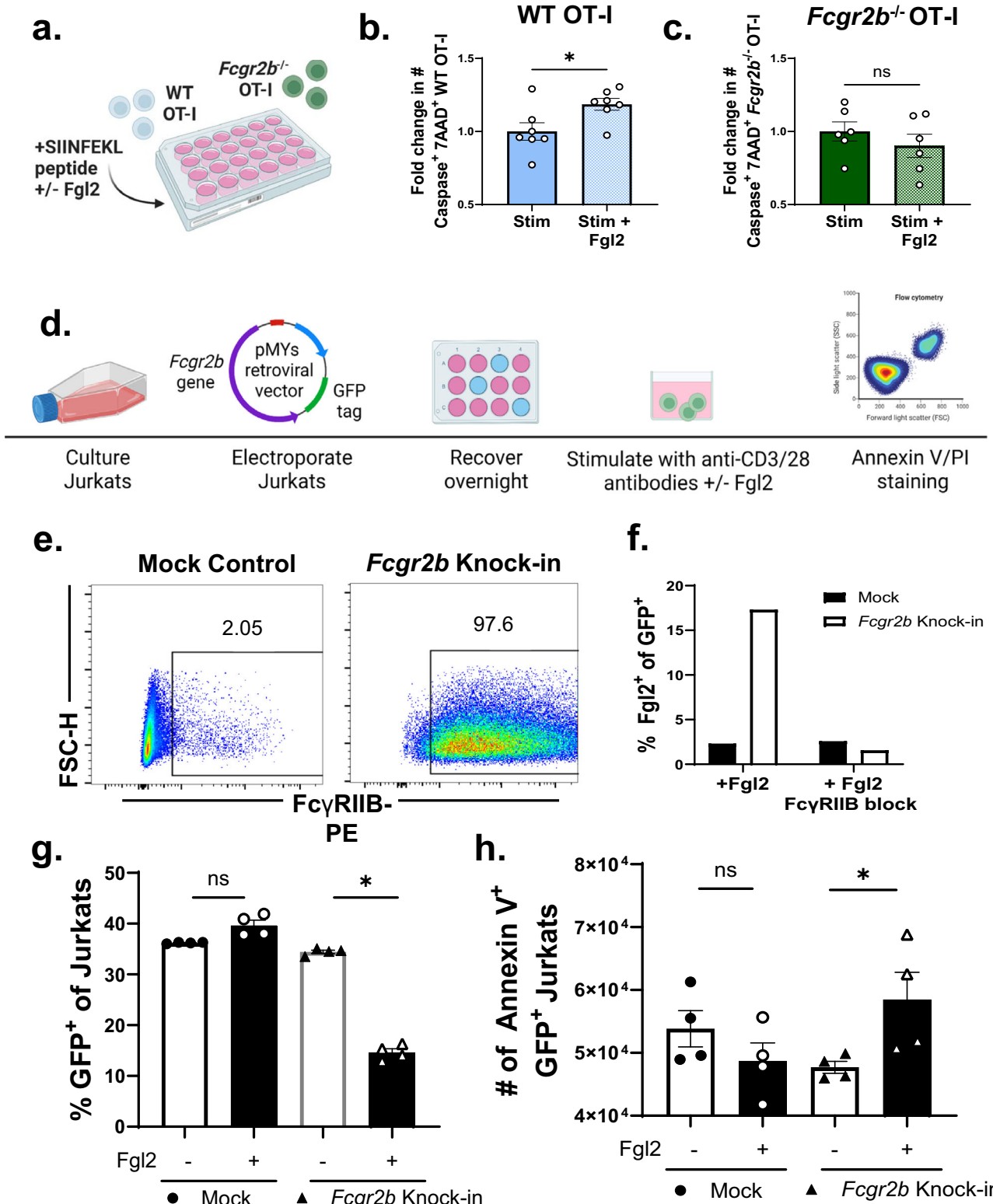

Dr. Mohamed Abdel-Hakeem. Mice were bled on days 4, 8, 12, 15 for flow cytometry. On day 15, mice were sacrificed and the spleen, blood, and lymph nodes of mice were harvested for flow cytometry.

**Murine cell processing, ex vivo peptide stimulation, and flow cytometry staining**

For tumor experiments, spleen, draining lymph node (right inguinal proximal to tumor), and tumors of mice were processed to cell suspensions, blood underwent red blood cell (RBC) lysis prior to staining.

For viral experiments, spleen, brachial and axillary lymph nodes were processed to cell suspensions, blood underwent RBC lysis prior to staining. The samples were then stained for surface markers prior to permeabilization for transcription factor staining using the antibodies listed in Supplementary Table 1. Cells were permeabilized using a FoxP3/transcription factor kit (Invitrogen). For Fgl2 cytokine staining of OT-I, splenocytes were ex vivo stimulated at 37 °C for 4 hours with 10 nM OVA$_{257\text{-}264}$ (SIINFEKL) peptide (B16-OVA) and 10 μg/mL Golgi-Plug (BD Biosciences). For Fgl2 cytokine staining of P14 cells,

**Fig. 5 | Fgl2 induces CD8⁺ T cell apoptosis via ligation of FcγRIIB in murine and human T cells. a** Schematic shown wherein WT or *Fcgr2b⁻/⁻* OT-I mouse splenocytes are stimulated with SIINFEKL peptide for three days, in the presence or absence of recombinant Fgl2 the last 24 h, cell death was then assessed via caspase 3/7 and 7AAD flow staining. Summary data of the caspase 3/7⁺/7AAD staining of **b** WT OT-I (*n* = 6 mice per group, *p* = 0.0379) vs. **c** *Fcgr2b⁻/⁻* OT-I (*n* = 7 mice per group) stimulated in the presence or absence of Fgl2, normalized to stimulated alone controls, pooled data from three independent experiments. **d** Schematic wherein the immortalized human Jurkat T cell line was transfected with a plasmid containing a GFP tag and human *Fcgr2b*. After recovery, transfected cells were serum starved and then stimulated with anti-CD3/28 antibody in the presence or absence of Fgl2. Cell death was assessed via Annexin V staining. **e** Representative flow plots showing FcγRIIB protein surface expression after transfection in mock-transfected vs. *Fcgr2b*-transfected cells. **f** Summary data showing Fgl2 staining on mock transfected or *Fcgr2b* transfected cells incubated with Fgl2 protein (*n* = 3 mice per group, average shown). In some wells, cells were pre-incubated with FcγRIIB block prior to Fgl2 incubation. Summary data showing **g** frequency of GFP⁺ (*p* = ns and *p* = 0.0285 respectively) and **h** cell number of Annexin V⁺ transfected mock or *Fcgr2b*-transfected Jurkats stimulated in the presence (open symbols) or absence (closed symbols) of Fgl2 (*p* = ns and *p* = 0.0285 respectively). Representative data from two independent experiments, *n* = 4 mice per group. Mann-Whitney non-parametric, unpaired two-sided tests were used were comparing two groups. Kruskall-Wallis non-parametric, one-way ANOVA with multiple comparisons was used when comparing >2 groups. The error bar in summary figures denotes mean ± SEM. **p* < 0.05. Source data are provided as a Source Data file. **a, d** is created with BioRender.com released under a Creative Commons Attribution-NonCommercial-NoDerivs 4.0 International license.

splenocytes were ex vivo stimulated at 37 °C for 4 hours with 0.4 μg/ml LCMV-gp₃₃₋₄₁ (KAVYNFATM) peptide and 10 μg/mL GolgiPlug (BD Biosciences). After 4 hours, cells were processed and stained for intracellular markers using antibodies listed in Supplementary Table 1. Fgl2 antibody (Abnova) were conjugated to fluorophore with Lightning Link technology (or isotype) and validated using *Fgl2⁻/⁻* splenocytes. The gating strategy followed is displayed in Supplementary Fig. 3a. All flow cytometry samples were acquired on a Fortessa or LSR II flow cytometer (BD Biosciences) and data were analyzed using FlowJo (Tree Star) and Prism (GraphPad Software). Absolute cell numbers were calculated using CountBright Beads (Life Technologies) according to the manufacturer's instructions.

### Plasmid design, cell line transfection, in vitro stimulation
Human *FCGR2B* gene was cloned into the pMYs-IRES-GFP expression vector, the plasmid without *Fcgr2b* gene was used as a mock control for experiments. The Jurkat E6-1 immortalized T cell line (ATCC) was cultured in RPMI 1640 (Sigma) and supplemented with 10% FBS, 1% P/ S, 1% HEPES, 1% L-glutamine, and 0.05 mM 2-ME. Cells were transfected with 5 μg of DNA of via electroporation according to Lonza manufacturer protocol. Cells were allowed to recover for 30 h, FcγRIIB surface staining and GFP expression were confirmed post-electroporation. Transfected cells were then stimulated with 10 μg/ ml of anti-CD3 (clone OKT3) and anti-CD28 (clone CD28.2) antibodies in serum starved media for 1 hour (Fig. 5f) or 24 h (Fig. 5g, h) in the presence or absence of 100 nM of Fgl2. Fgl2 binding was confirmed via flow staining with Fgl2-PE antibody (Abnova) and apoptosis was assessed with the Pacific Blue Annexin V/SYTOX AADvanced Apoptosis Kit (Thermofisher) used according to manufacturer instructions. Live/dead stain (Thermofisher) was also used to distinguish dead cells.

### In vitro stimulation of WT and Fcgr2b⁻/⁻ OT-I T cells in the presence of Fgl2
In total 5 × 10⁶ splenocytes isolated from WT or *Fcgr2b⁻/⁻* OT-I transgenic mice were stimulated with SIINFEKL peptide for 48 h. After 48 h, fresh peptide alone or fresh peptide and 1 μg/ml soluble Fgl2 (R&D Systems) was added. After 24 h, cells were harvested and caspase 3/7 and 7-AAD staining (Thermofisher) was assessed via flow cytometry.

### Human samples
The study design and conduct complied with the regulations on the use of human study participants and was conducted in accordance with the Declaration of Helsinki. This protocol was approved by Emory University's Institutional Review Board and all donors gave written informed consent for the collection and use of the samples. For TIL studies, melanoma tumor tissues were collected (IRB #00095411) deidentified, and distributed by the Cancer Tissue and Pathology shared resource of Winship Cancer Institute of Emory University.

These patient demographic data (n = 4) are shown in Supplementary Data Table 2.

### Patient tumor preparation and in vitro stimulation
For tumor-infiltrating lymphocyte (TIL) experiments, patient tumor tissue was collected and dissociated using the Human Tumor Dissociation Kit (Miltenyi) and GentleMACS Octo Dissociator (Miltenyi) according to the manufacturer's instructions. Cells were activated in vitro using 15 μl/ml anti-CD3/28-coated Dynabeads (Thermofisher) in R10 for 48 hours at 37 °C and 5% CO₂. Flow staining was performed using the antibodies listed in Supplementary Table 1. Fgl2 staining was assessed via staining with Fgl2-APC antibody conjugated with Lightning Link technology.

### Reanalysis of publicly available RNA-sequencing datasets
To investigate the presence of Fgl2 in the context of tumor, three independent single-cell RNA-sequencing data of murine B16 melanoma, human melanoma TIL, and human squamous cell carcinoma TIL were reanalyzed with the pipeline established in BBrowser2 (BioTuring)[26]. Carmona et al. [25] performed single cell RNA sequencing of B16 murine tumors excised 15 days post challenge, and CD8⁺ T cells were purified and sorted for sequencing. From seven samples, 7174 single-cell transcriptomes were obtained and the log of the normalized UMI counts + 1 was calculated. In the melanoma patient TIL dataset, Sade Feldman et al. [27] performed single cell RNA sequencing of 16,291 cells from 32 patients with melanoma utilizing SMART-Seq2. In the dataset deposited by Yost et al. [62], 32 squamous cell carcinoma patient tumors were digested and cells were FACS-sorted for single-cell RNA-sequencing, 79,046 cells passed quality control [16] Data plotting and statistical analysis were performed with GraphPad Prism 9. Datasets are available on the GEO database under accession numbers: GSE11639, GSE120575, and GSE123814 respectively.

We next investigated the presence of Fgl2 on antigen-specific CD8⁺ T cells during LCMV-Clone 13 in several published datasets. The single-cell RNA-sequencing dataset deposited by Miller et al. [31] was used to assess Fgl2 expression in both B16 and LCMV-Clone 13. Tetramer⁺ CD8⁺ T cells were isolated, sorted, and sequenced day 28 post LCMV infection and day 20 post B16 challenge, this dataset is available on the GEO database under accession number GSE122713. Other datasets from Wherry et al. (GSE9650)[9] Doering et al. (GSE41867)[7,34], and Hudson et al. [24], were also datamined. Unless otherwise noted, antigen-specific CD8⁺ T cells were isolated and sorted day 45 post LCMV-Clone 13 infection. The volcano plot of differentially expressed genes between exhausted and naive antigen-specific CD8⁺ T cells was generated using GEO2R from the Im et al. [23] dataset (GSE84105). The RNA sequencing dataset previously deposited by Morris et al. [16] was used to query *Fgl2* expression in FcγRIIB⁺ vs FcγRIIB⁻ CD8⁺ T cells (GSE118439).

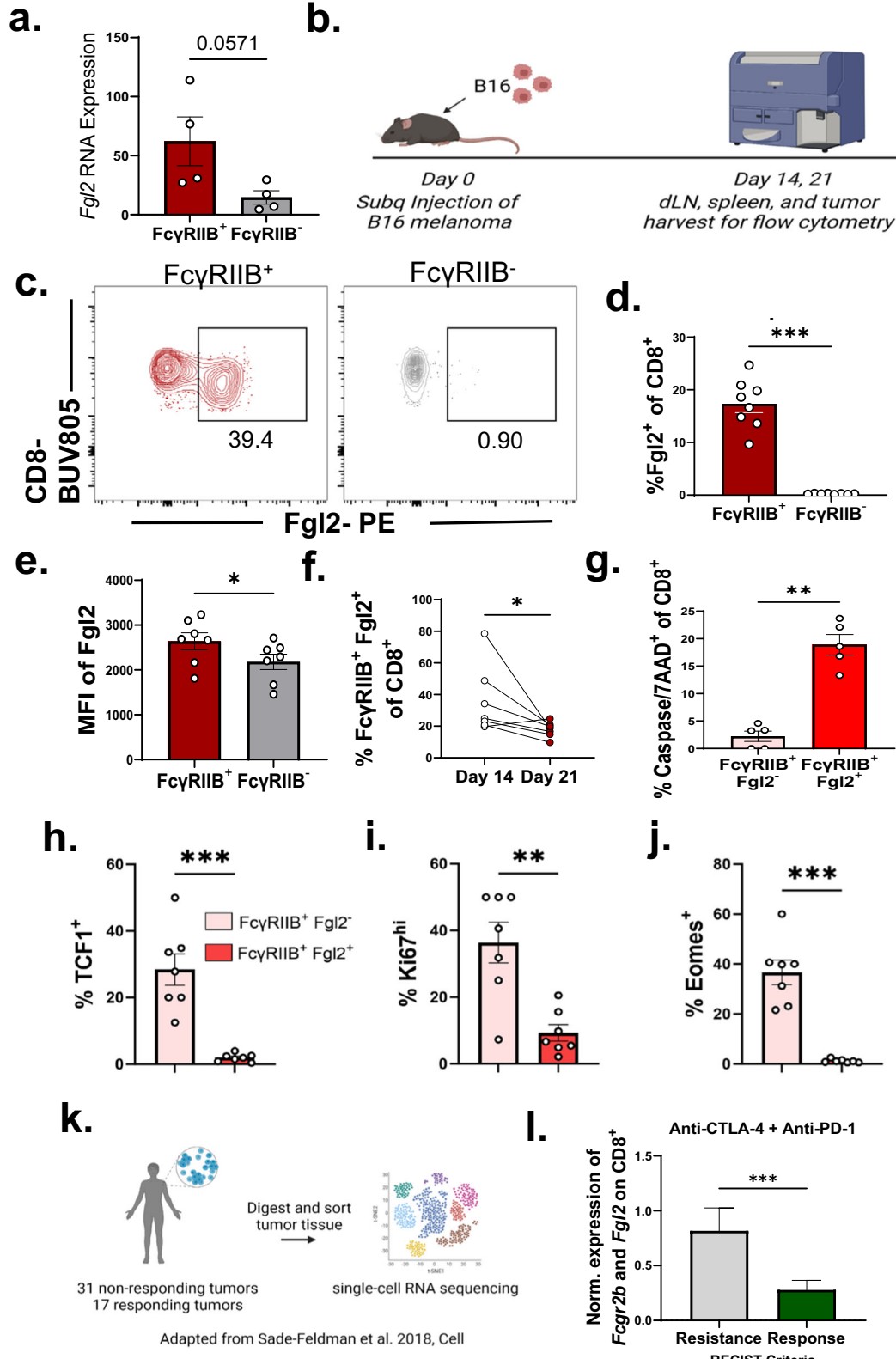

## Statistics

Non-parametric, unpaired t-test (Mann-Whitney) was used to compare cell populations between groups, while Wilcoxon matched-pairs rank tests were performed to compare subsets within the same donor. One-way ANOVA with multiple comparisons (Kruskal-Wallis) was performed when comparing multiple groups followed by a post-hoc Dunn's multiple comparison test. Tumor size and weight were calculated using a non-parametric, unpaired t-test (Mann-Whitney). The ROUT outliers test was used to determine any outliers. Samples sizes were determined by the resource equation model, 2-3 independent experiments were performed. All analyses were done using Prism (v 9.0, GraphPad Software). In all legends and figures, mean + SEM is shown, and $*p < 0.05$, $**p < 0.01$, $***p < 0.001$, $****p < 0.0001$.

**Fig. 6 | *Fgl2* is more expressed in FcγRIIB⁺ vs FcγRIIB⁻ CD8⁺ T cells, double positive cells are farther along T cell differentiation trajectory. a** Bar graph showing *Fgl2* RNA expression on CD44^hi FcγRIIB⁺ vs. CD44^hi FcγRIIB⁻ CD8⁺ T cells from previously published RNA sequencing data from Morris et al. (*n* = 4 per group)[16]. **b** Schematic showing B16 challenge in mice prior to sacrifice on days 14 and 21 for flow cytometry analyses. **c** Representative flow plots and summary data showing **d** frequency (*n* = 8 mice per group) and **e** mean fluorescence intensity (*n* = 7 mice per group) of Fgl2⁺ cells within FcγRIIB⁺ or FcγRIIB⁻ CD8⁺ T cells harvested from the spleen of B16-challenged mice (*p* = 0.0078 and *p* = 0.0312, respectively). **f** Summary data showing frequency of double positive FcγRIIB⁺Fgl2⁺ CD8⁺ T cells at day 14 vs day 21 post tumor challenge (*n* = 7 mice per group) (*p* = 0.0175). **g** Summary data showing caspase 3/7⁺ 7AAD⁺ CD8⁺ T cells within FcγRIIB⁺Fgl2⁻ vs. FcγRIIB⁺Fgl2⁺ CD8⁺ T cells from the spleen of B16-challenged mice (*n* = 5 mice per group, *p* = 0.0079). Bar graphs showing frequency of **h** TCF1⁺

(*p* = 0.0006), **i** Ki67^hi (*p* = 0.0041), and **j** Eomes⁺ (*p* = 0.0006) CD8⁺ T cells within FcγRIIB⁺Fgl2⁻ (pink bars) and FcγRIIB⁺Fgl2⁺ (red bars) CD8⁺ T cell populations from the draining lymph nodes of challenged mice, *n* = 7 per group. Representative data from two independent experiments. **k** Schematic and **l** bar graph comparing patient response (according to RECIST criteria) based on a signature of *Fgl2* and *Fcgr2b* expression on patient CD8⁺ TIL from a publicly available dataset deposited by Sade-Feldman et al. [5]. consisting of 16,291 single cell transcriptome profiles from patient tumors (*n* = 32, *p* = 0.0001), data was normalized to housekeeping gene expression within each cell is shown. Mann-Whitney non-parametric, unpaired two-sided test was used. The error bar, in summary, figures denotes mean ± SEM. *\*p* < 0.05 **\*\*p* < 0.01, \*\*\**p* < 0.001. **b** k was created with BioRender.com released under a Creative Commons Attribution-NonCommercial-NoDerivs 4.0 International license.

## Reporting summary

Further information on research design is available in the Nature Portfolio Reporting Summary linked to this article.

## Data availability

The data generated in this study have been deposited in the GEO database and is available under the GEO database accession number GSE252556 available here,). Source data are provided with this paper.

## Code availability

No custom software that is not publicly available was utilized in this study. However, code is available upon request.

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

## Acknowledgements

Next-generation sequencing services were provided by the Emory NPRC Genomics Core which is supported in part by NIH P51 OD011132. Sequencing data was acquired on an Illumina NovaSeq 6000 funded by NIH S10 OD026799. Research reported in this publication was sup-ported in part by the Cancer Tissue and Pathology shared resource of Winship Cancer Institute of Emory University and NIH/NCI under award number P30AC138292. Schematics in figures were created with Bior-ender.com. This study was supported in part by the Emory Flow Cyto-metry Core (EFCC), one of the Emory Integrated Core Facilities (EICF) and is subsidized by the Emory University School of Medicine. This study was supported by award AI164716 to M.L.F. K.B.B. was supported by NIH fellowships F31CA271764 and F99CA284255 during the study.

## Author contributions

K.B.B. designed, performed, and analyzed experiments and wrote the manuscript. D.L., A.S.D., and M.M.W. performed experiments. K.L.A. provided coding framework. M.A.H. and C.M.P. provided technical expertise and editing of the manuscript. M.L.F. designed experiments, provided funding, analyzed data, and wrote the manuscript.

## Competing interests

The authors declare no competing interests.
