## [Peer Review File · Nature Communications]

CD8+ T cell-derived Fgl2 regulates immunity in a cell-autonomous manner via ligation of FcγRIIBREVIEWER COMMENTS

Reviewer #1 (FcγRIIb, immunotherapy) (Remarks to the Author):

The manuscript by Bennion and colleagues reports on the role of Fgl2 as a regulator of T cell immunity. The topic is timely and of general interest for the wider scientific community. The current study builds on previous work by the group showing that a subset of cytotoxic T cells can express the inhibitory Fcγ-receptor FcRIIb. Apart from its ability to bind IgG immune complexes, FcRIIb was shown to bind Fgl2. As Fgl2-deficient mice phenocopy some effects of FcRIIb deficient mice, it was suggested that Fgl2 binding to FcRIIb mediates inhibitory, immunosuppressive effects. The current study demonstrates that in mouse tumor models as well as in human tumor patients Fgl2 may be involved in impairing/down-regulating cytotoxic T cell responses. Using mouse and human TIL from melanoma patients or B16F10 tumors the authors demonstrate that Fgl2 can be expressed in subsets of antigen experienced CD44^{hi} T cells. More importantly, Fgl2 expressing T cells also expressed PD-1, which is a marker for exhausted T cells. Interestingly, Fgl2 expression in PD-1⁺ CD8⁺ T cells in melanoma patients correlated with a decreased survival. By transferring Fgl2⁺ or Fgl2⁻ tumor specific T cells into Fgl2 deficient hosts, the authors demonstrate that Fgl2⁻ T cells are more active and the tumor size in these animals is smaller, indicative of a better tumor control via Fgl2⁻ T cells. Moreover, the study demonstrates that Fgl2⁺ exhausted T cells may undergo apoptosis. In a set of elegant T cell co-transfer studies the authors demonstrate the Fgl2 released from T cells regulates FcγRIIb expression on antigen-experienced T cells in an autonomous manner and induces apoptosis via caspase 3/7. Thus, a cell autonomous negative regulatory feedback loop was identified which may be critical for CD8⁺ T cell dependent control of virus infections and malignant tumor growth. The paper is written well and the data is presented in a clear and logic manner.

Points to be considered:

My only major point for further experimental data would be to use the experimental setting where Fgl2 sufficient and deficient T cells are transferred into hosts expressing Fgl2. Using Fgl2 deficient hosts creates a valid setting for an initial experiment, yet a very artificial one.

Reviewer #2 (CD8, anti-tumor) (Remarks to the Author):

This manuscript was a pleasure to review. Your identification of the Fgl2/FcγRIIB axis in Ag-specific T-cell exhaustion is well supported by your experimental models and data mining of publicly available data bases derived from human malignancies. My suggestions for revisions are few and relatively minor.

1. Fig 1f shows significant differences in normalized Fgl2 expression between PD-1/int and PD-1/high cells. But the corresponding bars look virtually identical. Please explain.
2. On p6 you write, "Together these data demonstrate that Fgl2 is produced by CD44^{hi} antigen-specific CD8⁺ T cells upon activation (~5%)." This was not clear; please elaborate.
3. On p7 you write, "Importantly, this pathway was not differentially expressed in the comparison analyses of naïve WT vs. Fgl2^{-/-} CD8⁺ T cells (data not shown)." This is

important enough to warrant showing the data so please provide it.

4. I was going to suggest you perform an additional experiment in which you blocked Flg2 with a mAb as a way to improve tumor efficacy in the melanoma model. But your citation of the glioblastoma model (ref #45) convinced me this was not necessary. However, I suggest you add a sentence or two noting the therapeutic clinical potential of this blocking approach.

Reviewer #1

The manuscript by Bennion and colleagues reports on the role of Fgl2 as a regulator of T cell immunity. The topic is timely and of general interest for the wider scientific community. The current study builds on previous work by the group showing that a subset of cytotoxic T cells can express the inhibitory Fcg-receptor FcRIIb. Apart from its ability to bind IgG immune complexes, FcRIIb was shown to bind Fgl2. As Fgl2-deficient mice phenocopy some effects of FcRIIb deficient mice, it was suggested that Fgl2 binding to FcRIIb mediates inhibitory, immunosuppressive effects. The current study demonstrates that in mouse tumor models as well as in human tumor patients Fgl2 may be involved in impairing/down-regulating cytotoxic T cell responses. Using mouse and human TIL from melanoma patients or B16F10 tumors the authors demonstrate that Fgl2 can be expressed in subsets of antigen experienced CD44^{hi} T cells. More importantly, Fgl2 expressing T cells also expressed PD-1, which is a marker for exhausted T cells. Interestingly, Fgl2 expression in PD-1⁺ CD8⁺ T cells in melanoma patients correlated with a decreased survival. By transferring Fgl2⁺ or Fgl2⁻ tumor specific T cells into Fgl2 deficient hosts, the authors demonstrate that Fgl2⁻ T cells are more active and the tumor size in these animals is smaller, indicative of a better tumor control via Fgl2⁻ T cells. Moreover, the study demonstrates that Fgl2⁺ exhausted T cells may undergo apoptosis. In a set of elegant T cell co-transfer studies the authors demonstrate the Fgl2 released from T cells regulates FcgRIIb expression on antigen-experienced T cells in an autonomous manner and induces apoptosis via caspase 3/7. Thus, a cell autonomous negative regulatory feedback loop was identified which may be critical for CD8⁺ T cell dependent control of virus infections and malignant tumor growth. The paper is written well and the data is presented in a clear and logic manner.

- We thank Reviewer 1 for their time in reviewing the manuscript and support of the work.

Points to be considered:

1. My only major point for further experimental data would be to use the experimental setting where Fgl2 sufficient and deficient T cells are transferred into hosts expressing Fgl2. Using Fgl2 deficient hosts creates a valid setting for an initial experiment, yet a very artificial one.

- We appreciate the reviewer's point and agree that further lines of investigation using WT recipients would be important in evaluating the contributions of Fgl2 from cell types other than CD8⁺ T cells. Future work in the lab will investigate potential alternative sources of Fgl2, and the role of these sources in anti-tumor immunity, and would therefore constitute a separate manuscript.
- We further agree that the model we used was a valid albeit reductionist approach to test the single variable of the contribution of Fgl2 from CD8⁺ T cells. We therefore submit that the data presented robustly support the conclusions made in the manuscript.

Reviewer #2

This manuscript was a pleasure to review. Your identification of the Fgl2/FcγRIIB axis in Ag-specific T-cell exhaustion is well supported by your experimental models and data mining of publicly available data bases derived from human malignancies. My suggestions for revisions are few and relatively minor.

We thank Reviewer 2 for their time in reviewing the manuscript and support of the work.

Points to be considered:

1. Fig 1f shows significant differences in normalized Fgl2 expression between PD-1^{int} and PD-1^{high} cells. But the corresponding bars look virtually identical. Please explain.

- Although visually small, the difference between the PD-1^{int} and PD-1^{hi} CD8⁺ T cell populations is statistically significant because the data depict 1249 individual naïve CD8⁺ T cells, 1548 PD-1^{int} effector CD8⁺ T cells, and 588 PD-1^{high} exhausted CD8⁺ T cells (i.e. the high n contributes to statistical significance). However, to address the reviewer's concern, we have modified the graph to better visually highlight the difference between the number of PD-1^{int} vs. PD-1^{hi} CD8⁺ T cells in Figure 1f. In addition, the numbers of individual cells analyzed each group is now included in Figure Legend 1 of the revised manuscript:

“t-SNE visualization and accompanying comparison of exhausted-like gene signature (Pcd1 and Havcr2) with Fgl2 expression on (G) naïve CD8⁺ T cells (n=1249 cells), (H) PD-1^{int} effector CD8⁺ (n=1548 cells), and (I) PD-1^{high} exhausted CD8⁺ T cells (n=588 cells).” (page 31, highlighted text)

- In addition, we have modified the text in the Statistics section of the Methods to provide additional detail on the post-hoc statistical tests used for this comparison:

“One-way ANOVA with multiple comparisons (Kruskal-Wallis) was performed when comparing multiple groups followed by a post-hoc Dunn's multiple comparison test.” (page 23, highlighted text)

2. On p6 you write, "Together these data demonstrate that Fgl2 is produced by CD44^{hi} antigen-specific CD8⁺ T cells upon activation (~5%)." This was not clear; please elaborate.

We apologize for the lack of clarity. To address this, we have added new text to the Results section expanding upon the experiments shown in Supplementary Figure 1e-k that support the conclusion that Fgl2 is produced by CD44^{hi} antigen-specific CD8⁺ T cells upon activation as copied below:

“Likewise, a significantly higher frequency and mean fluorescence intensity (MFI) of Fgl2 cytokine production was observed within CD44^{hi} CD8⁺ T cells compared to CD44^{lo} CD8⁺ T cells during ex vivo peptide stimulation of cells isolated from the spleen (p<0.01) and tumor (p<0.05) of B16-OVA challenged mice (Supplementary Fig. 1e-g). Additionally, we observed a significant increase in the MFI of Fgl2 and frequency of Fgl2 producers among OVA-specific CD8⁺ T cells during a multi-day stimulation with cognate antigen compared to

unstimulated controls ($p < 0.05$) (Supplementary Fig. 1h-k).” (page 5, highlighted text)

- In addition, the reference for each figure supporting the statement has been added to the summary paragraph in the revised manuscript for added clarity (page 6, highlighted text). The statement has also been revised to connect the two points more fluidly as copied below:

“Together these data demonstrate that *Fgl2* is produced by $CD44^{hi}$ antigen-specific $CD8^+$ T cells upon activation (~5%) (Supplementary Fig. 1), but that it is the $PD-1^+$ exhausted-like antigen-specific $CD8^+$ T cells that produce the most *Fgl2* (~40%) (Fig. 1).” (pages 6-7, highlighted text)

3. On p7 you write, "Importantly, this pathway was not differentially expressed in the comparison analyses of naïve WT vs. *Fgl2*^{-/-} $CD8^+$ T cells (data not shown)." This is important enough to warrant showing the data so please provide it.

- Unfortunately, the GSEA software only generates plots for pathways that *are* differentially enriched between the groups. Thus, within the naïve WT vs *Fgl2*^{-/-} $CD8^+$ T cell dataset, there is no GSEA plot generated by the software for the pathways “HALLMARK_APOPTOSIS” and “GSE41867_MEMORY_VS_EXHAUSTED_CD8_TCELL_DAY30_LCMV_UP” because they are not differentially enriched between the groups. In this case, we submit that text alone is sufficient to make the point that those two GSEA pathways are absent from the list of differentially expressed pathways in naïve WT vs *Fgl2*^{-/-} $CD8^+$ T cells, and that the best approach is to remove the phrase “data not shown” from the text, because there is no data to show.

For the reviewer’s reference, screenshots pathways that are differentially expressed (cut off p value = 0.05) in naïve WT vs. naïve *Fgl2*^{-/-} $CD8^+$ T cells are shown below. Additionally, we have revised the text as copied below:

“Of note, these pathways were not differentially expressed in the comparison analyses of naïve WT vs. *Fgl2*^{-/-} $CD8^+$ T cells.” (page 7-8, highlighted text)

Hallmark gene sets

ID <chr>	Description <chr>	setSize <int>	enrichmentScore <dbl>	NES <dbl>
HALLMARK_INFLAMMATORY_RESPONSE	HALLMARK_INFLAMMATORY_RESPONSE	157	0.6662716	2.096681
HALLMARK_ALLOGRAFT_REJECTION	HALLMARK_ALLOGRAFT_REJECTION	168	0.6412157	2.022980
HALLMARK_KRAS_SIGNALING_UP	HALLMARK_KRAS_SIGNALING_UP	142	0.6194823	1.926123
HALLMARK_COMPLEMENT	HALLMARK_COMPLEMENT	155	0.5636494	1.775105
HALLMARK_ESTROGEN_RESPONSE_LATE	HALLMARK_ESTROGEN_RESPONSE_LATE	140	0.5572079	1.730455
HALLMARK_GLYCOLYSIS	HALLMARK_GLYCOLYSIS	157	0.5488137	1.727054

6 rows | 1-6 of 11 columns

Hallmark C7 gene sets

ID <chr>	Description <chr>
GSE11057_PBM_C_VS_MEM_CD4_TCELL_UP	GSE11057_PBM_C_VS_MEM_CD4_TCELL_UP
GSE10325_LUPUS_CD4_TCELL_VS_LUPUS_MYELOID_DN	GSE10325_LUPUS_CD4_TCELL_VS_LUPUS_MYELOID_DN
HOEK_NK_CELL_2011_2012_TIV_3D_VS_ODY_ADULT_3D_DN	HOEK_NK_CELL_2011_2012_TIV_3D_VS_ODY_ADULT_3D_DN
GSE10325_BCELL_VS_MYELOID_DN	GSE10325_BCELL_VS_MYELOID_DN
GSE22886_NAIVE_CD8_TCELL_VS_MONOCYTE_DN	GSE22886_NAIVE_CD8_TCELL_VS_MONOCYTE_DN
GSE22886_NAIVE_BCELL_VS_NEUTROPHIL_DN	GSE22886_NAIVE_BCELL_VS_NEUTROPHIL_DN

6 rows | 1-2 of 11 columns

4. I was going to suggest you perform an additional experiment in which you blocked Flg2 with a mAb as a way to improve tumor efficacy in the melanoma model. But your citation of the glioblastoma model (ref #45) convinced me this was not necessary. However, I suggest you add a sentence or two noting the therapeutic clinical potential of this blocking approach.

- We thank you for the helpful feedback. The following sentences describing the potential clinical promise and considerations of using an anti-Fgl2 blocking antibody in melanoma have been added to the revised manuscript as copied below:

“This work extends the work of other groups that have shown that Fgl2 has a protumor role in the context of glioma and hepatocellular carcinoma^{29,30,40,44-46}. However, previous studies have focused on the impact of Fgl2 produced from regulatory T cells, macrophages and tumor-associated cells (stroma, fibroblasts) as these cell types are known cellular sources of Fgl2 that can mediate immunosuppression on T cells^{28-30,32,33,40-42,45-49}. As such, these studies showed the therapeutic efficacy of antibodies blocking Fgl2, suggesting that this approach may hold clinical promise in melanoma.” (pages 15-16, highlighted text)

REVIEWERS' COMMENTS

Reviewer #1 (Remarks to the Author):

The authors have chosen not to address my major concern. This is somewhat dissapointing as all the experimental methods are established and this is a very straightforward experiment. Without this data set the title and claims of the study are too broad and should be adjusted at the least.

Reviewer #2 (Remarks to the Author):

Thank you for addressing my prior critique. Congratulations on your noteworthy research.